

# Soil organic carbon stocks did not change after 130 years of afforestation on a former Swiss Alpine pasture

Tatjana C. Speckert[1], Jeannine Suremann[1], Konstantin Gavazov[2], Maria J. Santos[1], Frank Hagedorn[2], Guido L. B. Wiesenberg[1]

[1]Department of Geography, University of Zurich, Winterthurerstrasse 190, 8057 Zurich
[2] Swiss Federal Institute for Forest, Snow and Landscape Research, Zürcherstrasse 111, 8903 Birmensdorf

*Correspondence to*: Tatjana C. Speckert (tatjanacarina.speckert@geo.uzh.ch)

**Abstract.** Soil organic matter (SOM) plays an important role in the global carbon cycle, especially in alpine ecosystems. However, ongoing forest expansion in high elevation systems potentially alters SOM storage through changes in organic matter (OM) inputs and microclimate. In this study we investigated the effects of an *Picea abies* L. afforestation chrono-sequence (40-130 years) of a former subalpine pasture in Switzerland on soil organic carbon (SOC) stocks and SOM dynamics. We found that SOC stocks remained relatively constant throughout the chrono-sequence, with comparable SOC stocks in the mineral soils after afforestation and previous pasture ($SOC_{40}$-year-old forest= 11.6± 1.1 kg m$^{-2}$, $SOC_{130}$-year-old forest=11.0 ± 0.3 kg m$^{-2}$, and SOC pasture= 11.5 ± 0.5 kg m$^{-2}$). However, including the additional carbon of the organic horizons in the forest, reaching up to 1.7 kg m$^{-2}$ in the 55-year-old forest, resulted in a slight increase in overall SOC stocks following afforestation. We found that the soil C:N ratio in the mineral soil increased in the topsoil (0-5cm) with increasing forest stand age, from 11.9 ± 1.3 in the grassland to 14.3 ± 1.8 in the 130-year-old forest. In turn, we observed a decrease in soil C:N ratio with increasing depth in all forest stand ages. This suggests that litter-derived organic matter (C:N from 35.1 ± 1.9 to 42.4 ± 10.8) is likely incorporated and translocated from the organic horizon to the mineral topsoil (0-10 cm) of the profiles. As roots had very high C:N ratios (pasture 63.5 ± 2.8 and forests between 54.7 ± 3.9 and 61.2 ± 2.9), particulate root-derived organic matter seems to have a minor influence on forest soil C:N ratio and thereby on SOC stock accumulation in the mineral soil. These results suggest that, although the afforestation only moderately affected the SOC stock, there is an apparent alteration in the SOC dynamics through changes of the litter composition caused by the vegetation shift. We conclude that spruce afforestation on a former subalpine pasture does not necessarily change the total SOC stock and that consequently there is no SOC sequestration on a decadal to centennial scale.

## 1 Introduction

Soil organic matter (SOM) plays an important role in the global carbon cycle, and it is essential for soil fertility and nutrient availability (Prietzel et al., 2016). Furthermore, it increases soil stability and consequently reduces the risk of soil erosion (Garcia-Pausas et al., 2017). Especially in mountain ecosystems, SOM is of particular importance, as it forms thick organic layers (Pizzeghello et al., 2017), and contributes significantly to slope stability (Djukic et al., 2010). In alpine environments,





cold temperatures result in slower decomposition and lead to more labile particulate SOM compared to temperate soils (Hagedorn et al., 2019; Garcia-Pausas et al., 2017). Specifically, alpine grasslands contain proportionally more easily decomposable plant residues compared with temperate soils (Zimmermann et al., 2007). Moreover, the chemical characteristics of these plant residues still resemble the ones of the original organic matter (OM) input. This makes alpine SOM extremely

sensitive to climate warming (Hagedorn et al., 2010). With global warming, accelerated decomposition of SOM will lead to SOM losses, which increases $CO_2$ emissions from soil that enhance the feedback on climate change (Prietzel et al., 2016, Soong et al., 2021). Afforestation of former pastures is one promising measure of mitigating rising atmospheric $CO_2$ levels and consequently also global warming, as it typically contributes to carbon sequestration in biomass (Bastin et al., 2019). In the European Alps, the conversion of pastures to forests is the dominant land-use change due to land abandonment

(Zimmermann et al., 2010), resulting in an expansion of the forest area by one third during the last 150 years (Brändli, 2020). Afforestation of pastures directly affects SOM dynamics through alteration of OM inputs, as well as through the quality of root and plant litter, and changes in soil properties (De Deyn et al., 2008; Hiltbrunner et al., 2013). In pastures, the OM input usually occurs through root turnover and rhizodeposition (Solly et al., 2013). In forests, however, OM accumulates in living biomass, as well as in the organic horizons (Peichl et al., 2012). While the increase in aboveground OM following afforestation

is widely demonstrated (Thuille and Schulze, 2006; Risch et al., 2008; Guidi et al., 2014; Smal et al., 2019), the effects on belowground OM are more controversial (Hong et al., 2020). Numerous studies reported contrasting results concerning carbon sequestration following afforestation including decrease of SOC stocks (Guo et al., 2007; Risch et al., 2008), initial decrease followed by increase of SOC stocks after decades (Thuille and Schulze, 2006), increase of SOC stocks (Grünzweig et al., 2007; Popleau and Don, 2013), and no effects on the SOC stocks (Davis et al., 2007). These diverging trends likely depend on

environmental and soil properties, tree species as well as forest age (De Gryze et al., 2004; Guo and Gifford, 2002; Paul et al., 2002). One key factor for the SOC stock changes appears to be the initial carbon stock. Therefore, an increase often occurs in SOC-poor soils, while a decrease happens in SOC-rich soils (Hong et al.,2020). However, to date most studies have focused on the upper soil horizons (e.g., 0-5 cm, Pérez-Cruzado et al., 2014) leaving gaps in our understanding of what happens at greater soil depths. For example, a decrease in SOC stock was observed following afforestation of previously SOC-rich soils

in the deep soil layers with the largest decrease between 10-20 cm depth in an afforested area in Northern China (Hong et al., 2020). Also forest age affects SOM dynamics by directly altering the litter input and its decomposition due to less decomposable compounds with an increased C:N ratio (Gunina et al.,2017). Therefore, it could be expected that longer afforestation periods might have different effects on SOM dynamics than shorter ones. This remains an underexplored aspect as frequently relatively young afforestation sequences have been studied that typically range between 30 and 50 years (Guo et

al., 2007; Grünzweig et al., 2007; Strand et al., 2021). Tree species composition of forests can also have effects on SOM dynamics, as for example the SOC stock and SOC accumulation in the organic horizons is generally higher under coniferous forest compared to mixed-deciduous forests (Gosheva et al., 2017). Organic horizons are typically rich in OM that has not been stabilized by interactions with minerals and they are thus more responsive to environmental changes compared to mineral soils (Yanai et al., 2003). Further, afforestation using coniferous trees on a former pasture result in a considerable change of



the rooting system with lower fine root biomass, and higher root C:N ratio and lignin content (Hiltbrunner et al., 2013) compared to pasture. This can result in a lower belowground carbon input into mineral soil. Pastures, on the other hand, are characterized by a higher fine root biomass (< 2 mm) with higher turnover rates and may thus provide a greater C input into SOC than forests (Solly et al., 2013).

The objectives of this study were to investigate SOC stocks and the long-term carbon sequestration potential of an afforestation

chrono-sequence of Norway spruce (*Picea abies* L.) on a former pasture in the Swiss Alps. The specific aims were i) to examine whether SOC and nitrogen stocks changed in relation to different forest stand ages (40- 130 years), ii) to identify the potential sources of SOM, and iii) to identify whether C and N dynamics are affected by litter input and quality of forest stand ages (40 – 130 years). We hypothesized that with increasing forest stand age after pasture conversion there is i) an increase in the SOC stock due to the changes in litter input and quality towards higher C:N ratios and less easily decomposable OM (e.g.,

less herbaceous leaves and more needles and woody tissues). Further we expect ii) a shift from more root-derived carbon input in the pasture towards more litter-derived carbon input in the forest sites and iii) an increase in SOC accumulation in the organic horizons with increasing forest stand age.

## 2 Material and Methods

### 2.1 Study site

Our study was conducted on a south-exposed slope located above the village of Jaun in the Canton of Fribourg, Switzerland [7°15'54 E; 46°37'17 N] at an altitude between 1450 and 1600 meters above sea level. The soil was classified according to WRB (2014) as a Leptic Eutric Cambisol Clayic for both, pasture (clay: 60 %, silt: 30 %, sand: 8 %) and forest areas (clay: 50 %, silt: 35 %, sand: 12 %). Mean air temperature is 11.4 °C in summer and 0.6°C in winter with a mean annual precipitation of 1250 mm (Hiltbrunner et al., 2013). The pasture has been used for cattle grazing in the past (Hiltbrunner et al., 2013). After

severe avalanches in 1954 and 1968 (municipality of Jaun, 2021), the pasture was gradually afforested with Norway spruce (*Picea abies* L.). This afforestation process lasted several decades and resulted in stand ages between 40 and 55 years. An old forest stand (>130 years) has been covered by trees for a much longer period as assessed by tree rings and some strongly decomposed stumps. However, the exact stand age for this old forest could not be accessed, which is the reason, why we call this a 130-year-old forest. All tree ages were estimated based on tree diameter and tree rings and verified by aerial photographs

and historical maps. The plant community on pasture soils mainly consisted of herbaceous species with ribgrass (*Plantago lanceolata* L.) and reed fescue (*Festuca arundinacea* Schreb.) as dominant species. Norway spruce (*Picea abies* L.) was the dominant species in all forest stand ages.

### 2.2 Sampling design

In our project, we focus on four different forest ages of the afforestation sequence: Pasture (0-year-old forest) as control and

forest stand ages of 40 years, 55 years, and 130 years. These were chosen because of a moderate elevation gradient and





comparable soil properties among and within the different forest stand ages (Hiltbrunner et al., 2013), which allows to investigate changes in SOC composition over several decades. Further, our choice was supported by field inspections, including terrain and soil properties. For every forest stand age, five individual plots with a size of 10 m x 10 m [=100 m2, which are 1 A = Ar] were distributed along an elevational gradient over the entire area of the respective forest and pasture, respectively (Fig. 1). The size of the individual forest stand ages ranges from ca. 70 A (130-year-old) to ca. 130 A (55-year-old).


### 2.3 Sampling and sample preparation

The sampling campaign was conducted in July 2020. Forest canopy density and the number of trees was determined in the field for the five plots for all forest stand ages. The number of dead and living spruce trees as well as the number of deciduous trees in the same area were recorded in relation to forest age (Table S1). Material of organic soil horizons (O-horizons) was collected in July 2020 on three plots in the 40-, 55-, and in the 130-year-old forest stands (N = 27; n = 3). The samples of the O-horizons were separated into Oi (slightly decomposed organic material), Oe (moderately decomposed organic material), and Oa (highly decomposed organic material; Jahn et al., 2006) for all plots and their thickness was measured during field work. O-horizons were lacking in the pasture area and therefore not collected and measured. Five soil pits in the pasture area (0-year-old) as well as three soil pits for each forest stand age (40-, 55-, and 130-year-old) were prepared with dimensions of at least 100 cm width x 50 cm depth. The slope ranged between 25-30° at all sites. Within the soil pits, slope-parallel levels were prepared, and roots were counted on these levels using a grid of 50 cm x 50 cm, according to Gocke et al. (2016). In addition to the counting on slope-parallel levels, roots were counted on three profile walls using the same grid as above to identify the possible sources of SOM as well as to get an overview of the 3-dimensional distribution and variation of root abundance in the soil. The mineral soil samples (N = 118) were taken with two volumetric steel cylinders (100 cm$^3$) on these slope-parallel levels that were incrementally increased by 5 cm to a maximum depth of 45 cm (pasture n = 5; forest areas n = 3). Bulk density was determined in every 5cm interval using the volume of the steel cylinders (100 m$^3$) and the mass of the dried soil (Table S1). All soil samples were stored in open plastic bags until their arrival in the laboratory, where they were stored at -20°C. O-horizon material was freeze-dried and weighed until constant weight. Afterwards, the O-horizon material was separated into residues of pinecones, twigs, needles, leaves, moss, lichens, mycorrhiza, and other organisms. Mineral soil samples were oven dried (40 °C) until constant weight and passed through a 2 mm sieve to remove any stones, roots, and plant residues (Ofiti et al., 2021). Roots (N = 180) were afterwards manually removed with tweezers and separated into fine (0-2 mm; N = 92) and coarse (2-5 mm, N = 88) roots (pasture n = 5; forest n = 3). Soil particles that were attached to the roots were removed by washing the roots with deionized water (Solly et al., 2013) in a sonication bath (Richter-Heitmann et al., 2016). Roots were afterwards dried at 40°C until constant weight (Solly et al., 2013).










## 2.4 Laboratory analysis

A subsample of all dried samples was ground in a ball mill (MM400, Retsch, Haan, Germany). The milled and homogenized mineral soil sample were acidified with HCl to remove carbonates, washed with deionized water, and afterwards dried in the oven at 40°C (Volk et al., 2018). Results of elemental analysis were corrected for mass loss. Dried root samples were combined
(0-5 mm) due to low amounts of the fine root samples. Material of the roots, organic horizons, and mineral soil samples before and after carbonate removal were analysed in duplicate for carbon (C) and nitrogen (N) concentrations as well as stable carbon isotope ($\delta^{13}$C) composition using a Thermo Fisher Scientific Flash HT Elemental Analyser, coupled to a Delta V Plus isotope ratio mass spectrometer via ConFlo IV (Thermo Fisher Scientific, Bre-men, Germany). The calibration was performed with caffeine (Merck, Germany), and a soil reference sample originated from a haplic Chernozem soil (clay: 19 %, fine sand: 53
%; Harsum, northern Germany; University of Zurich, 2023). The results of the $\delta^{13}$C values are presented in per mil (‰) relative to the Vienna Pee Dee Belemnite (V-PDB) standard. For the mineral soil samples, total nitrogen (TN concentrations and stocks) was derived from the measurements without carbonate removal, while soil organic carbon (SOC concentrations and stocks) was obtained from the samples after carbonate removal.

## 2.5 Data analysis

Total SOC and TN stocks per unit area [kg m$^{-2}$] were calculated per individual plot. SOC and TN stocks per individual depth interval (0.05m) were calculated using the soil thickness D of the horizon [m], its bulk density $\rho$ [g cm$^{-3}$], its SOC concentration C [%], and 10 as a unit con-version factor Eq. (1):

$$X[kg\ m^{-2}] = (C * D * \rho) * 10 ,\qquad\qquad\qquad(1)$$

where X represents the SOC and TN stocks [kg m$^{-2}$], respectively, of the individual depth interval. The total SOC and TN
stocks of a given depth (0-45 cm) were calculated as follows Eq. (2):

$$\sum_{i=1}^{h} Xi \qquad\qquad\qquad(2)$$

whereby h is the number of the 5cm depth interval from 0 to 45 cm (h = 9, except for one plot in the pasture where h = 7). We then calculated the average ± standard error (SE) for each stand age for SOC and TN stocks as well as $\delta^{13}$C values of mineral soil, organic horizons, and roots, respectively. We then tested whether SOC, TN, and $\delta^{13}$C values differed between land use
practices (pasture or forest) or among individual stand ages using a one-way analysis of variance (ANOVA, $p < 0.05$) followed by a post-hoc Tukey HSD test (p adj < 0.95). To test the effect of afforestation on the SOM dynamics, we used forest age as predictor and soil depth as a fixed effect of SOC in each depth interval, specifying as a random effect the distinct plots in the stand age classes in order to reflect the grouped sampling within soil pits. These were inputs to a set of mixed-effect models. We tested model fit with REML method using the nlme package and setting the significance level at 0.05. Data analysis was
performed with R software 4.0.4 (R Core Team, 2020).



## 3. Results

### 3. 1 Vegetation composition

The number of living spruce trees was significantly higher in the 40-year-old forest (forest$_{40}$ = 44 ± 10 A$^{-1}$, forest$_{55}$ = 18 ± 2 A$^{-1}$, forest$_{130}$ = 13 ± 1 A$^{-1}$) than in other forest stand ages (F$_{(2, 12)}$ = 9.55, p = 0.003; Table S1). Further, we also found that the 55-year-old forest had a significantly higher number of dead spruce trees (forest$_{55}$ = 6 ± 1 A$^{-1}$, forest$_{40}$ = 1 ± 1 A$^{-1}$ , forest$_{130}$ = 2 ± 1 A$^{-1}$) than the other stands (F$_{(2, 12)}$ = 4.94, p = 0.027). The 130-year-old forest showed a significantly higher number of broadleaf tree species (forest$_{130}$ = 8 ± 4 A$^{-1}$, forest$_{55}$ = 4 ± 1 A$^{-1}$, forest$_{40}$ = 1A ± 1 A$^{-1}$) than the other stand ages (F$_{(2, 12)}$ = 2.44, p = 0.129). The highest (65 ± 5 %) canopy cover was observed in the 130-year-old forest, followed by the 55-year-old (48 ± 3 %) and 40-year-old forest (58 ± 5 %), but these were not significantly different (F$_{(2, 12)}$ = 3.24, p = 0.075).

### 3. 2 Macroscopic organic matter remains in upper (Oi) organic horizons

In all forest stand ages, pinecones, and branches were significantly more abundant in the Oi-horizon (F$_{(5, 8)}$ = 9.161, p = 0.004; Table S2) compared with Oe- and Oa-horizons. The highest proportion of cone residues was found in the 55-year-old forest (65 mass - %), followed by the 130-year-old (58 mass - %) and 40-year-old forests (33 mass - %). Also needles made up almost one fourth (25 %) of the mass  in Oi-horizons of the 40-year-old forest but only around 10 % in the 55-year-old (5 %) and 130-year-old forest (8  %).

### 3. 3 Root biomass and frequency

We found significant differences in the fine root biomass (0-2 mm) between pasture and the different forest ages (F$_{(3, 88)}$ = 5.21, p = 0.002). Highest fine root biomass (0-45 cm) was found in the pasture (184.9 ± 39.5 g m$^{-2}$) and lowest fine root biomass was observed in the 55-year-old forest (2.9 ± 1.4 g m$^{-2}$; Fig. 2a). A post-hoc Tukey HSD test showed no significant differences (p adj < 0.95) in the fine root biomass in between afforested areas, but a significant difference between pasture and the 40-year-old and 130-year-old forests, respectively. In contrast, there was no difference in the coarse root biomass (2-5mm) between pasture and the different forest stand ages (F$_{(3, 84)}$ = 1.29, p = 0.282; Fig. 2b). Highest coarse root biomass was found in the 130-year-old forest (683.5 ± 202.1 g m$^{-2}$) and lowest in pasture areas (274.2 ± 209.2 g m$^{-2}$). Both, fine (F$_{(1,7)}$ = 11.15, p = 0.012) and coarse root biomass (F$_{(1, 7)}$ = 30.41, p = 0.001) significantly decreased with increasing soil depth (Table S3). This is in line with the root frequency that was counted on horizontal levels and profile walls, where we observed the highest fine root frequency (~15 800 m$^{-2}$) in the pasture and the highest coarse root frequency in the 40-year-old forest (~2500 m$^{-2}$; Table S3) and decreasing numbers with depth.

### 3. 4 Soil organic C and N stocks

As organic matter horizons were completely missing in the grassland during the sampling campaign, organic horizons could be only investigated in afforested areas. The combined SOC stock of the organic horizons (Oi, Oe, and Oa) was highest in the





55-year-old forest (1.7 ± 0.2 kg m$^{-2}$) followed by the 130-year-old (1.3 ± 0.2 kg m$^{-2}$) and the 40-year-old forest (0.8 ± 0.1 kg m$^{-2}$; $F_{(1,25)}$ = 0.065, p = 0.801; Fig. 3a). The post doc Tukey HSD test showed significant differences (p adj <0.95) in the combined SOC stock of the organic horizons between the 55-year-old and the 40-year-old forest. Highest SOC stock was found in the Oi horizon in the 55-year-old (0.9 ± 0.1 kg m$^{-2}$) with a drop in the SOC stock from Oi to Oe in all forest stand

ages (Table 1). The total SOC stock of the mineral soil (0-45 cm) in the 40-year-old (11.6 ± 1.1 kg m$^{-2}$) and the 130-year-old forest (11.0 ± 0.3 kg m$^{-2}$) was almost identical as in the pasture soil (11.5 ± 0.5 kg m$^{-2}$). In contrast, the total SOC stock of the mineral soil (0-45 cm) in the 55-year-old forest was significantly lower compared to the soils of the pasture as well as the other forest stand ages (8.3 ± 0.6 kg m$^{-2}$; $F_{(3, 114)}$ = 4.02, p = 0.009). If the organic horizons are considered, there was an increase in carbon in the 40-year-old (+7 %) and 130-year-old forest (+ 11 %) as compared to the pasture soils, but this increase was not

significant ($F_{(2, 14)}$ = 1.64, p = 0.229). Within the soil profiles, highest SOC stocks occurred in the topsoil (0-5; 5-10 cm) in both, pasture and forest soils, and decreased with increasing soil depth ($F_{(1,7)}$ = 71.44, p < 0.001; Table 1). The TN stock of the organic horizons (Oi, Oe, Oa) was highest in the 55-year-old forest (0.05 ± 0.01 kg m$^{-2}$) followed by 130-year-old and 40-year-old forest, but these differences were not significant ($F_{(1, 25)}$ = 0.157, p = 0.696; Fig. 3b). The TN stock of the mineral soil (0-45 cm) of the pasture (1.16 ± 0.08 kg m$^{-2}$) equals the one in the 40-year-old forest (1.09 ± 0.08 kg m$^{-2}$), followed by the

130-year-old (0.98 ± 0.02 kg m$^{-2}$) and 55-year-old forest (0.85 ± 0.06 kg m$^{-2}$; $F_{(3, 114)}$ = 5.31, p = 0.002). Similar like the SOC stock, also the TN stock significantly ($F_{(1, 7)}$ = 85.62, p = 0.001) decreased with depth (Table 1).

### 3. 5 SOM composition

The C:N ratio of the O-horizons (Oi, Oe, Oa) in the forest was not significantly different be-tween the different forest ages ($F_{(1, 25)}$ = 2.134, p = 0.157; Table 3). In all forest stand ages, highest C:N ratios were observed in the Oi-horizons (forest$_{130}$ =

42.4 ± 10.8, forest$_{55}$ = 37.5 ± 3.1, forest$_{40}$ = 35.5 ± 1.9), which consistently decreased from Oi- to Oa-horizons.
The C:N ratio of roots (0-5 mm diameter) did not significantly differ ($F_{(3, 98)}$ = 1.55, p = 0.208) between pasture and afforested areas (Table 2). Highest root C:N ratio was observed in the pasture (0-5 cm) soil with 65.3 ± 6.1 followed by the 130-year-old (59.9 ± 3.9), the 40-year-old (50.9 ± 4.6), and the 55-year-old forest (43.0 ± 3.8).
Within the O-horizons the δ$^{13}$C values increased from Oi towards Oa in the 40-year-old ($F_{(2, 6)}$ = 2.389, p = 0.173) and the 55-

year-old forest ($F_{(2, 6)}$ = 12.15, p = 0.008). In contrast, the δ$^{13}$C values did not change significantly from Oi- (-27.4 ± 0.2 ‰) towards Oa-horizons (-27.0 ± 0.2 ‰) ($F_{(2, 6)}$ = 0.212, p = 0.815) in the 130-year-old forest. Within the mineral soil, the δ$^{13}$C values increased ($F_{(3, 114)}$ = 12.04, p < 0.001) with increasing forest age with highest differences pronounced in the subsoil (40-45 cm) ranging between -23.3 ± 2.4 ‰ in the 40-year-old forest and -25.5 ± 0.2 ‰ in the pasture site (Table 3). In all soil profiles, the δ$^{13}$C values increased with increasing soil depth. Pasture roots had δ$^{13}$C values that varied between -27.8 ± 0.2 ‰

and -27.1 ± 0.1 ‰ with depth. These values were significantly different from roots ($F_{(3, 98)}$ = 11.80, p < 0.001), recovered from afforested areas (forest$_{40}$ = between -26.9 ± 0.3 ‰ and -25.9 ‰; forest$_{55}$ = between -27.3 ± 0.7 ‰ and -26.1 ‰; forest$_{130}$ = between -27.6 ± 0.2 and -26.9 ± 0.4 ‰). Considering the soil depth of 45 cm, the δ$^{13}$C values of the roots became more (~ +1 ‰) positive with increasing soil depth in the pasture as well as in afforested areas ($F_{(1, 7)}$ = 1.88, p = 0.213; Table 2).





## 4 Discussion

### 4.1 SOC stock changes depending on forest age

We measured a SOC stock of 11.5 ±0.5 kg m$^{-2}$ in the subalpine pasture (Fig. 3a), which is in line with the SOC stock observed by Hiltbrunner et al., (2013) with a SOC stock of 13.4 ± 1.2 kg m$^{-2}$ in alpine pasture. One possible reason for this minor difference in the SOC stock between the current study and Hiltbrunner et al. (2013) could be the difference in sampling depth of 0-60 cm in Hiltbrunner et al., (2013) and 0-45 cm in this study.  In contrast, our results are higher than those of other studies, which reported SOC stocks between 4 – 6 kg m$^{-2}$ in subalpine grassland (Zeemann et al., 2010) in Switzerland. The SOC stock of forest areas in this study is within the range of 8.3 ± 0.6 kg m$^{-2}$ (55-year-old forest) to 11.6 ± 1.1 kg m$^{-2}$ (40-year-old forest) and is in line with the reported SOC stocks for Swiss forest soils (Nussbaum et al. 2014) ranging between1-36 kg m$^{-2}$ (median 6.3 kg m$^{-2}$). Our findings that reported an increase of the SOC stock in the mineral soil after 40 years of afforestation is conforming to other studies which reported an increase in the SOC stock after 40 to 50 years of afforestation (Thuille and Schulze, 2006, Hiltbrunner et al. 2013). In our study, the 40-year-old forest is not only the youngest in the investigated chrono-sequence, but it experienced some transversal clear cuts a few years ago as part of forest management. Consequently, the higher SOC stock in the 40-year-old forest as compared to older stands can further be explained by the presence of grass roots. They typically dominate the carbon input in grasslands (Thuille and Schulze 2006) and generate additional SOC input compared to the 55-year-old and 130-year-old forest (Laganière et al., 2010), where the grasses disappeared. In the same line, the 40-year-old forest showed an almost identical TN stock as the pasture (Fig. 3b), which suggests that only a small amount of nitrogen sequestered by trees.  The decrease of the SOC stock in the 55-year-old compared to the 40-year-old forest agrees with other chrono-sequences (Post and Kwon, 2000; Thuille and Schulze, 2006; Poepleau et al., 2011). One possible explanation could be the absence of a grass cover as reported by Hiltbrunner et al. (2013), which is also supported by the lower fine- and coarse-root biomass in the 55-year-old forest found in this study compared to the 40-year-old forest. Further, avalanche protection infrastructure was built only 10-15 m uphill from one of our profiles, which might have resulted in potential soil disturbance resulting from avalanche construction or former avalanches in that area. The cause of the observed effects cannot be entirely elucidated to-date, but all of these factors could have caused a loss of carbon compared with the other forest stands. Especially the latter explanation might align with the SOC stock in relation to soil depth, where we observed a significantly lower SOC stock at 15 to 25 cm soil depth, compared to 40-year-old and 130-year-old forest. If the 55-year-old forest is excluded from the analysis, then there is no significant difference ($F_{(2, 89)}$ = 0.40, p = 0.669) in the SOC stock in the subsoil (30 - 45 cm) after afforestation. In addition, there is no significant difference in the SOC stock reported by Hiltbrunner et al, (2013), ten years ago in the same study area, and the reported SOC stock in this study. Therefore, there is no increase in the SOC stock with increasing forest age. To better understand why we have no increased C sequestration, even after decades of afforestation, the use of molecular proxies would be one possibility to identify potential sources of OM as well as to identify the alteration in its composition (Jansen and Wiesenberg, 2017).



In addition to the carbon stored in the mineral soil, all forest soils showed well differentiated organic horizons, where additional carbon has been accumulated. Consequently, a carbon sequestration of an additional $0.8 \pm 0.1$ and $1.7 \pm 0.2$ kg m$^{-2}$ occurred in the organic horizons of afforested areas compared to the pasture (Fig. 3a) at Jaun. This agrees with other studies reporting the potential of an additional carbon sequestration in organic horizons of 2.3 kg m$^{-2}$ (Hiltbrunner et al., 2013) and of 2.5 kg m$^{-2}$

(Thuille and Schulze, 2006) following afforestation. In contrast to Hiltbrunner et al. (2013), we did not find a gradual increase in the SOC stock of the organic horizons with increasing forest age at the same site. This can be due to the sampled locations and erosion of the unconsolidated organic horizons that might have occurred at selected areas due to snow melt or heavy rainfall events. The higher SOC stock in organic horizons of the 55-year-old forest compared to the 40-year-old and 130-year-old forest reported in the current study may be explained by the litter composition, which mainly consisted of dead wood

residues (Table S1). This material is less decomposable than leaf or needle litter, which resulted in a lower decomposition (Cortrufo et al., 2013) and higher carbon accumulation and therefore thicker organic horizons (3.2 cm $\pm$ 0.4 cm) compared to the 40-year-old and 130-year-old forest stands. Overall, if the organic horizons are combined with the mineral soil, then there was an increased SOC sequestration within 40-, and 130 years of afforestation. This confirms our hypothesis of an increase in the SOC sequestration caused by an additional carbon accumulation on the organic horizons. However, this way of carbon

accumulation has to be regarded with caution as organic matter found in organic horizons in steep alpine areas is much less protected than mineral bound SOC (Thuille and Schulze, 2006). This argues for an increased sensitivity of SOC stocks in forested alpine areas and asks for further protection measures, e.g., during forest renewal, where specifically organic horizons can be exposed to erosion and degradation (Bettoni et al., 2023).

**4.2 SOC alteration vs. sequestration with forest age**

While mineral SOC stock tended to remain unchanged within the investigated afforestation sequence and the SOC stock increased within the first 55 years, it remained an open question, to which extent mineral SOM contains preserved carbon from the pasture or how this is replaced by forest-derived carbon. The incorporation of forest-derived carbon can occur either via litterfall and the continuous incorporation and degradation of litter within the organic horizons on the one hand and via root organic matter on the other hand. With increasing forest age, the C:N ratio of the Oi horizons (Table 1) increased, which argues

for a lower decomposability of the litter caused by an increase in woody litter with increasing stand age (Schulp et al., 2008). This shift in the litter composition towards less decomposable material was previously described (Pérez-Cruzado et al., 2014; Strand et al., 2021). Furthermore, higher C:N ratio in the organic horizons as a function of tree age was also reported by Thuille and Schulze (2006) with an increase in woody litter such as twigs, branches, and pinecones. In addition, element concentration, such as nitrogen, decrease with increasing needle age (Linder, 1995), which could be another reason for the observed shift

towards less decomposable material. With ongoing decomposition from Oi to Oa horizons, the C:N ratio decreased, which was confirmed in all forest ages at Jaun.

On the one hand, the C:N ratio in the mineral soil increased as a function of forest age resulting in the highest C:N ratio of $14.3 \pm 1.8$ in the 130-year-old forest (Table 3) and on the other hand, the C:N ratio decreased with soil depth, specifically in





the top 20 cm. This demonstrates that litter-derived organic matter was likely incorporated and translocated from litter to
mineral soil in the top 20 cm of the profiles, while partially very low C:N ratios in deeper soils argue for an increased
degradation of organic matter in forest soils (Lorenz et al., 2020). As roots had very high C:N ratios, particulate root-derived
organic matter seems to have no significant influence on forest soil organic matter composition in the mineral soil, whereas it
cannot be excluded that root exudates  (Jílková et al., 2022) and root-associated mycorrhiza might have caused these values
(Lindahl et al., 2006). Consequently, aboveground litter quantity and quality seems to have the biggest effect on SOM dynamic
in the topsoil (Hiltbrunner et al., 2013). This is in line with previous findings, where a large proportion of the carbon input in
afforested areas is mainly through litterfall (Hiltbrunner et al., 2013, Bárcena et al., 2014), whereas this can be restricted to the
top 20 cm in our study.  Based on the $\delta^{13}C$ values, a shift from between -25.5 ± 0.2 ‰ to -26.9 ± 0.3 ‰  in the original pasture
to between -24.4 ± 0.3 ‰ to -26.0 ± 0.2 ‰ in the 130-year-old forest (Table 1). This indicates a re-placement towards tree-
derived carbon as a main source for the SOM in afforested areas. The enrichment in $^{13}C$ with increasing forest age could be
indicative for a strong enrichment of the heavier $^{13}C$ isotope during decomposition. As forest roots are more depleted in $^{13}C$ (-
1 to 3 ‰) compared to the mineral soil, it seems that roots do not have a significant effect on the $\delta^{13}C$ values of the mineral
soil. In contrast, fungal species, especially the saprotrophic fungi, are known to be enriched in $^{13}C$ (-22.8 ± 0.3 ‰; Hobbie et
al., 2001) and thus could be a potential explanation for the observed $^{13}C$ enrichment in the forest soil, especially in the 130-
year-old forest (Table 3). Along with the study of Hiltbrunner et al., (2013), we also observed a high variability between the
$\delta^{13}C$ values of the roots and the mineral in the oldest (130-year-old) forest stand age which is another argument for a slower
root turnover with increasing forest age. Additionally, the mineral soil in the pasture areas is less enriched in $^{13}C$, which is also
the case for the pasture root $\delta^{13}C$ values, compared to roots from the forest. Also, this suggests fine roots as major C source in
the pasture areas (Solly et al., 2013; Hiltbrunner et al., 2013).

## 5 Conclusion

Natural forest encroachment into abandoned alpine meadows and a rise in the treeline have led to a forest expansion of
European alpine areas during the past decades. In our study, we investigated the effects of afforestation on a former pasture on
the total SOC stock as well as on the possible alteration of the SOM dynamics. Conclusively, the SOC stock of the mineral
soil did not change 40 and 130 years, respectively, after afforestation compared with the original pasture soil. If the organic
horizons are considered, there was an increase in carbon stocks by 1-2 kg m⁻². This lack of additional sequestration in mineral
soil C and the restriction of C sequestration specifically in organic horizons poses limitation to the suitability of afforestation
measures for climate change mitigation. Specifically, organic horizons are highly vulnerable to erosion in steep alpine areas
as well as to losses upon disturbances and therefore the restriction to C sequestration in organic horizons is less sustainable
than in mineral soil. Although, the effects on the SOC stock following afforestation are only moderate in the investigated area,
the vegetation shift resulted in an obvious alteration in the SOC dynamics through changes in the litter composition and changes
in root-derived organic matter. The alteration towards a higher C:N ratio with increasing forest age, especially in the Oi horizon
reflects this alteration in less easily decomposable compounds. Specifically, the $\delta^{13}C$ values and C:N ratios strongly suggest





an alteration of soil organic matter composition with increasing forest age likely resulting from changes of organic matter sources and its degradation without allowing final conclusions. To better understand the processes following forest expansion by changes in land-use and climate in alpine areas, further investigations are needed to specifically quantify the sources of
organic matter and SOM degradation under different land-use in alpine areas.

**Authors contribution**

TCS: Conceptualization; methodology; writing and editing – original draft; investigation; data curation; formal analysis. JS: Investigation, review and editing. KG: Conceptualization; resources; validation; review and editing. MJS: data curation; formal analysis; review and editing. FH: resources; validation, review and editing. GLBW: Conceptualization; methodology; funding;
resources; review and editing.

**Competing interests**

The contact author has declared that none of the authors has any competing interests.

**Acknowledgements**

We thank Silvan Wick, Yves Brügger and Carrie L. Thomas for helping during field work. We further thank Jeannine Suremann, Aline Hobie, Barbara Siegfried, Dmitry Tichomirov, and Esmail Taghizadeh for their support during lab work. We acknowledge funding by the Swiss National Science Foundation (SNSF) under contract 188684 of the IQ-SASS project (Improved Quantitative Source Assessment of organic matter in Soils and Sediments using molecular markers and inverse modelling) to GLBW and project number PZ00P2_174047 to KG.

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

**Figure caption**

**Figure 1: Study area and sampling locations color-coded: black for  pasture;  yellow for the 40-year-old forest; white for the 55-year-old forest; and purple for the 130-year-old forest. (Map was originated with QGIS 3.22.5)**

**Figure 2: a) Fine and b) coarse (average± SE) root biomass (0-45cm) in the mineral soil of pasture and forest stands.**

**Figure 3 a) Soil organic carbon stocks and b) total nitrogen stocks in the O-horizons and the mineral soil (0-45cm; average ± SE) by vegetation cover and forest age.**

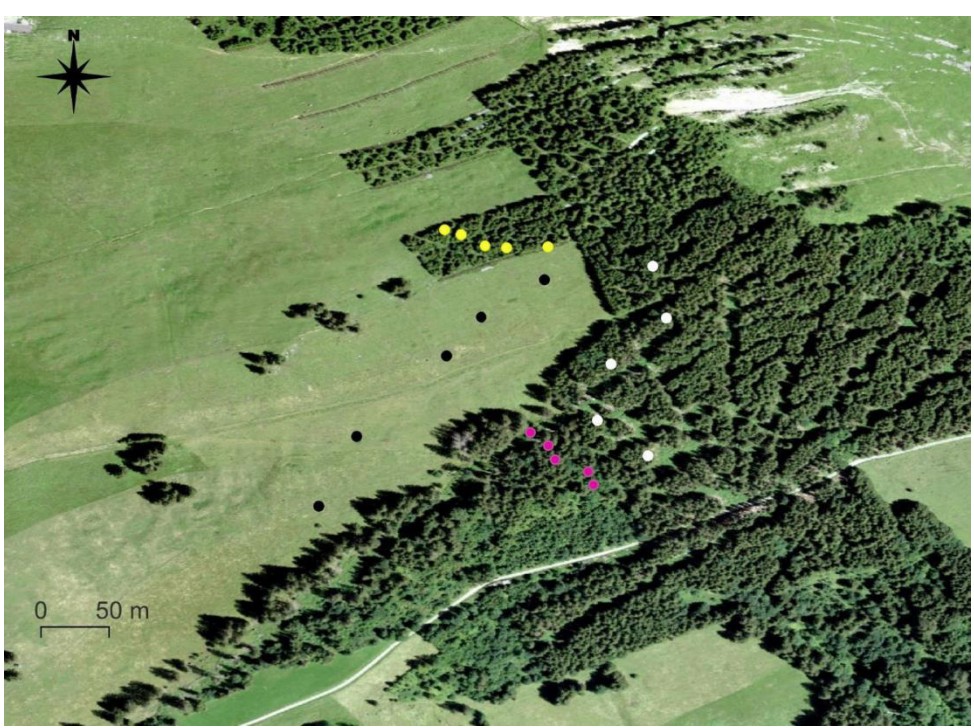

**Figure 1**






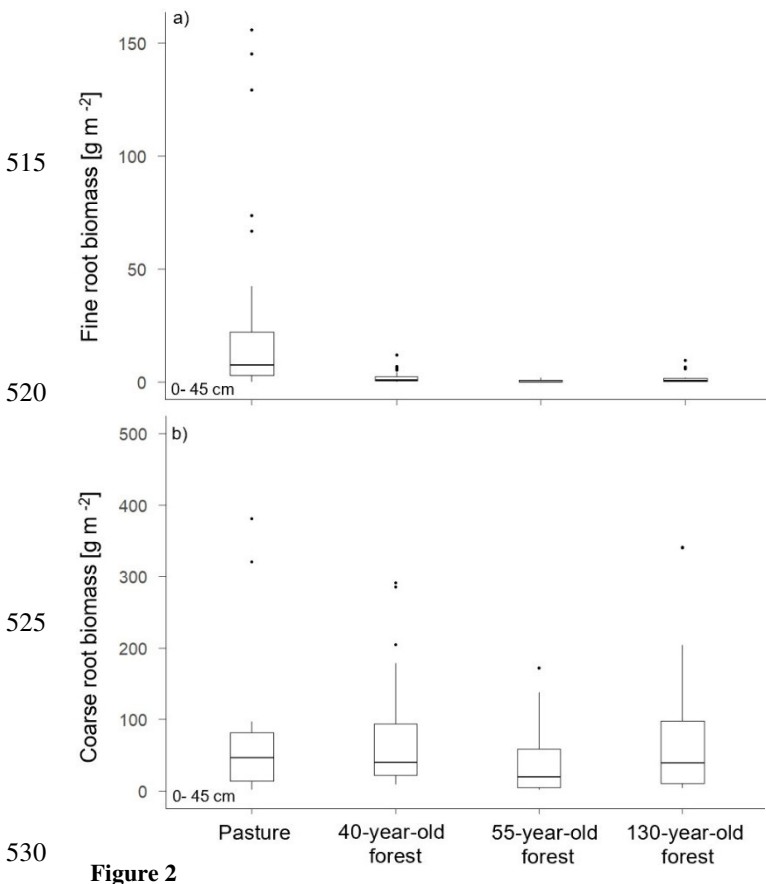

**Figure 2**



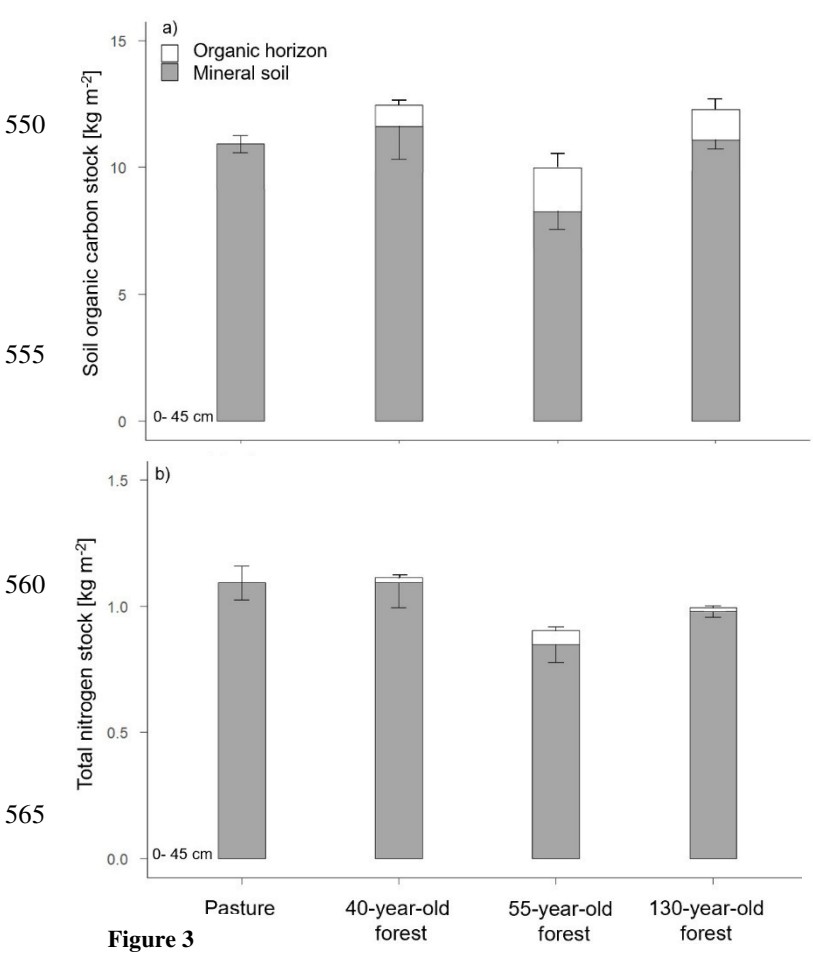

**Figure 3**



Table 1: Soil organic carbon, total nitrogen stocks of O-horizons (n=3), mineral soil (n=5 for pasture, n=3 for forest). Values are average ± SE (n.a. = not available, * Sum ± SE (0-45cm)).

| | Soil organic carbon [kg m$^{-2}$] | | | | Total nitrogen [kg m$^{-2}$] | | | |
|---|---|---|---|---|---|---|---|---|
| | | Forest | | | | Forest | | |
| O-horizons | Pasture | 40-year-old | 55-year-old | 130-year-old | Pasture | 40-year-old | 55-year-old | 130-year-old |
| Oi | n.a. | 0.3 ± 0.0 | 0.9 ± 0.1 | 0.7 ± 0.2 | n.a. | 0.01 ± 0.00 | 0.02 ± 0.00 | 0.02 ± 0.00 |
| Oe | n.a. | 0.4 ± 0.0 | 0.5 ± 0.1 | 0.3 ± 0.0 | n.a. | 0.02 ± 0.00 | 0.02 ± 0.00 | 0.01 ± 0.00 |
| Oa | n.a. | 0.1 ± 0.0 | 0.4 ± 0.3 | 0.3 ± 0.2 | n.a. | 0.01 ± 0.00 | 0.02 ± 0.01 | 0.01 ± 0.00 |
| *Sum | n.a. | 0.8 ± 0.1 | 1.7 ± 0.2 | 1.3 ± 0.2 | n.a. | 0.03 ± 0.00 | 0.05 ± 0.01 | 0.04 ± 0.01 |
| Mineral soil | | | | | | | | |
| 0-5cm | 2.2 ± 0.1 | 1.9 ± 0.2 | 1.5 ± 0.5 | 1.7 ± 0.3 | 0.19 ± 0.02 | 0.14 ± 0.02 | 0.11 ± 0.02 | 0.15 ± 0.01 |
| 5-10cm | 1.9 ± 0.1 | 1.7 ± 0.2 | 1.7 ± 0.5 | 1.9 ± 0.1 | 0.19 ± 0.01 | 0.13 ± 0.03 | 0.11 ± 0.03 | 0.14 ± 0.00 |
| 10-15cm | 1.4 ± 0.0 | 1.7 ± 0.2 | 0.8 ± 0.3 | 1.5 ± 0.1 | 0.15 ± 0.01 | 0.16 ± 0.01 | 0.12 ± 0.02 | 0.13 ± 0.00 |
| 15-20cm | 1.1 ± 0.1 | 1.5 ± 0.2 | 0.7 ± 0.2 | 1.5 ± 0.1 | 0.13 ± 0.01 | 0.14 ± 0.01 | 0.11 ± 0.01 | 0.13 ± 0.00 |
| 20-25cm | 1.2 ± 0.2 | 1.2 ± 0.1 | 0.9 ± 0.2 | 1.1 ± 0.0 | 0.13 ± 0.02 | 0.12 ± 0.01 | 0.10 ± 0.00 | 0.11 ± 0.00 |
| 25-30cm | 1.2 ± 0.2 | 1.4 ± 0.2 | 0.9 ± 0.2 | 1.2 ± 0.1 | 0.12 ± 0.02 | 0.14 ± 0.01 | 0.09 ± 0.01 | 0.11 ± 0.00 |
| 30-35cm | 1.1 ± 0.2 | 1.1 ± 0.1 | 0.8 ± 0.1 | 0.9 ± 0.1 | 0.10 ± 0.02 | 0.12 ± 0.01 | 0.09 ± 0.01 | 0.09 ± 0.01 |
| 35-40cm | 0.8 ± 0.1 | 0.9 ± 0.1 | 0.6 ± 0.0 | 0.9 ± 0.0 | 0.08 ± 0.01 | 0.11 ± 0.00 | 0.07 ± 0.00 | 0.08 ± 0.01 |
| 40-45cm | 0.7 ± 0.1 | 0.9 ± 0.1 | 0.7 ± 0.1 | 0.6 ± 0.1 | 0.07 ± 0.01 | 0.09 ± 0.00 | 0.07 ± 0.01 | 0.05 ± 0.03 |
| *Sum | 11.5 ± 0.5 | 11.6 ± 1.1 | 8.3 ± 0.6 | 11.0 ± 0.3 | 1.16 ± 0.08 | 1.09 ± 0.08 | 0.85 ± 0.06 | 0.98 ± 0.02 |

Table 2. C:N ratio and δ$^{13}$C values of root (0-5mm) samples (n= 5 for pasture; n= 3 for forest).Values are average ± SE (n.a. = not available, ** Average ± SE of the individual plots; n= 5 for pasture; n= 3 for forest).

| | C:N ratio | | | | δ$^{13}$C [‰ V-PDB] | | | |
|---|---|---|---|---|---|---|---|---|
| | | Forest | | | | Forest | | |
| Roots [0-5mm] | Pasture | 40-year-old | 55-year-old | 130-year-old | Pasture | 40-year-old | 55-year-old | 130-year-old |
| 0-5cm | 65.3 ± 6.1 | 50.9 ± 4.6 | 43.0 ± 3.8 | 59.9 ± 3.9 | -27.8 ± 0.2 | -26.6 ± 0.1 | -26.9 ± 0.2 | -27.2 ± 0.1 |
| 5-10cm | 57.5 ± 5.8 | 43.9 ± 4.7 | 65.8 ± 15.5 | 57.4 ± 8.6 | -27.2 ± 0.3 | -26.9 ± 0.3 | -26.5 ± 0.2 | -27.5 ± 0.2 |
| 10-15cm | 69.4 ± 1.3 | 52.9 ± 7.5 | 49.8 ± 4.9 | 68.5 ± 11.5 | -27.5 ± 0.2 | -26.5 ± 0.1 | -27.3 ± 0.7 | -27.6 ± 0.2 |
| 15-20cm | 62.6 ± 7.1 | 54.3 ± 5.5 | 63.2 ± 11.5 | 60.9 ± 3.5 | -27.1 ± 0.1 | -26.8 ± 0.3 | -26.8 ± 0.3 | -27.1 ± 0.2 |
| 20-25cm | n.a. | 67.2 ± 1.8 | 39.9 | 52.0 ± 3.8 | n.a. | -26.6 ± 0.4 | -27.6 | -27.3 ± 0.0 |
| 25-30cm | 65.6 | 71.1 ± 9.2 | 54.7 ± 2.0 | 51.8 ± 6.4 | -27.9 ± 0.0 | -26.4 ± 0.3 | -26.6 ± 0.5 | -27.3 ± 0.2 |
| 30-35cm | 61.0 ± 18.7 | 71.3 ± 4.9 | n.a. | 50.6 ± 8.4 | -27.4 | -26.6 ± 0.2 | n.a. | -27.5 ± 0.5 |
| 35-40cm | 69.7 | 56.6 ± 0.3 | n.a. | 59.6 ± 13.2 | -27.8 | -26.6 ± 0.6 | n.a. | -27.2 ± 0.5 |
| 40-45cm | n.a. | 75.6 | 76.1 | 76.6 ± 4.6 | n.a. | -25.9 | -26.1 | -26.9 ± 0.4 |
| **Average | 63.5 ± 02.8 | 57.1 ± 2.6 | 54.7 ± 3.9 | 61.2 ± 2.9 | | | | |





Table 3. $\delta^{13}$C values and C:N ratio of O-horizon samples (n=3), mineral soil samples (n= 5 for pasture; n= 3 for forest).
Values are average ± SE ((n.a. = not available; ** Average ± SE of the individual plots; n= 5 for pasture; n= 3 for forest).

| | C:N ratio | | | | $\delta^{13}$C [‰ V-PDB] | | | |
|---|---|---|---|---|---|---|---|---|
| | | Forest | | | | Forest | | |
| O-horizons | Pasture | 40-year-old | 55-year-old | 130-year-old | Pasture | 40-year-old | 55-year-old | 130-year-old |
| Oi | n.a. | 35.5 ± 1.9 | 37.5 ± 3.1 | 42.4 ± 10.8 | n.a. | -27.3 ± 0.1 | -27.7 ± 0.1 | -27-3 ± 0.4 |
| Oe | n.a. | 24.9 ± 1.9 | 26.6 ± 2.4 | 26.6 ± 4.2 | n.a. | -27.1 ± 0.3 | -27.1 ± 0.3 | -27.5 ± 0.3 |
| Oa | n.a. | 17.5 ± 0.9 | 14.8 ± 0.9 | 28.8 ± 6.7 | n.a. | -26.5 ± 0.3 | -26.3 ± 0.2 | -27.2 ± 0.3 |
| **Average | n.a. | 25.9 ± 2.7 | 26.3 ± 3.5 | 32.6 ± 4.6 | n.a. | | | |
| **Mineral soil** | | | | | | | | |
| 0-5cm | 11.9 ± 1.3 | 12.9 ± 0.5 | 13.1 ± 1.4 | 14.3 ± 1.8 | -26.9 ± 0.3 | -26.8 ± 0.2 | -26.2 ± 0.2 | -26.0 ± 0.2 |
| 5-10cm | 10.1 ± 0.9 | 13.7 ± 1.6 | 16.6 ± 4.6 | 13.6 ± 0.6 | -26.8 ± 0.2 | -26.5 ± 0.2 | -25.8 ± 0.2 | -25.6 ± 0.2 |
| 10-15cm | 9.3 ± 0.5 | 10.7 ± 1.1 | 6.8 ± 1.7 | 11.4 ± 0.6 | -26.3 ± 0.2 | -25.7 ± 0.2 | -25.6 ± 0.1 | -25.3 ± 0.1 |
| 15-20cm | 9.0 ± 1.3 | 10.3 ± 0.8 | 6.3 ± 1.8 | 11.1 ± 0.6 | -26.3 ± 0.1 | -25.9 ± 0.1 | -25.5 ± 0.2 | -24.9 ± 0.1 |
| 20-25cm | 9.8 ± 0.6 | 10.2 ± 0.9 | 8.2 ± 1.8 | 9.6 ± 0.2 | -25.9 ± 0.1 | -25.7 ± 0.1 | -25.6 ± 0.0 | -24.6 ± 0.2 |
| 25-30cm | 9.5 ± 0.3 | 9.4 ± 1.3 | 10.1 ± 2.1 | 10.9 ± 0.7 | -25.9 ± 0.1 | -25.7 ± 0.0 | -25.5 ± 0.2 | -24.9 ± 0.0 |
| 30-35cm | 10.4 ± 0.8 | 9.5 ± 0.9 | 8.5 ± 0.4 | 9.9 ± 0.8 | -25.5 ± 0.1 | -25.4 ± 0.2 | -25.4 ± 0.4 | -24.4 ± 0.3 |
| 35-40cm | 10.1 ± 0.6 | 8.8 ± 1.2 | 8.2 ± 0.7 | 10.1 ± 0.5 | -25.6 ± 0.1 | -24.7 ± 1.2 | -25.2 ± 0.2 | -24.6 ± 0.2 |
| 40-45cm | 10.7 ± 0.9 | 8.9 ± 1.6 | 8.8 ± 1.3 | 10.5 ± 0.6 | -25.5 ± 0.2 | -23.3 ± 2.4 | -25.1 ± 0.2 | -24.6 ± 0.0 |
| **Average | 10.2 ± 0.3 | 10.6 ± 0.5 | 9.7 ± 0.9 | 11.3 ± 0.4 | | | | |