# Peer review of "Soil organic carbon stocks did not change after 130 years of afforestation on a former Swiss Alpine pasture"

_EGUsphere, 2023_

## Referee Comment (RC2)

Speckert et al. Soil organic carbon stocks did not change after 130 years of afforestation on a former Swiss Alpine pasture

**Review notes**

The study presented here covers a topic that is interesting and relevant since many pastures in alpine regions are abandoned and expected to become forests in the future – which could feed back to climate change. Knowing the impact of this land-use conversion is needed. The topic and proposed research questions however are not very novel and do not contribute much new knowledge.

There are two things I value in the current manuscript:

- The long gradient in time (if the chronosequence would have been replicated) adding more information on the long term impacts of afforestation
- The study is well executed (- the design) and the manuscript is well written

My major concern:

- The study is based on pseudoreplications so the conclusions in terms of time since afforestation are not supported by the data, they could be coincidence or site effects. In my opinion this is a major issue that is hard to resolve.

Specific comments:

| Line | Comment |
|---|---|
| 19-22 | As roots … -> unclear sentence |
| 22-25 | Is this novel enough? |
| 37-38 | Afforestation is a promising measure -> is this also true for alpine regions where albedo effects might have a negative impact on climate change |
| 49-50 | Be explicit and call it "context-dependency" |
| 51 | Initial carbon stock is a key factor -> yet you do not take this into account in your research |
| 54-55 | Transition line 54 to 55 not very fluent (10-20 cm is not that deep? Was that the point?) |
| 58-59 | Age is underexplored -> I think this is the main strength of your study that you have a large gradient in time going back 130 years |
| 75-77 | Clear hypothesis. Is (iii) really needed? It is accepted as common knowledge I would say |
| Introduction | Overall very well written introduction with nice literature review. |
| 85 | Afforested -> Encroached? Natural regeneration? Planted to avoid erosion? Not explicitly stated in the M&M. If planted, at what planting distances? What is was the basal area (the density) of the stands at the time of sampling. This is important information that is needed to interpret the results |
| M&M | There are no replications of the different ages. The replications are made within the same forest stands resulting in pseudoreplications. This means you cannot say anything about the effect of age / time since afforestation. It can just be differences in between individual stands that do not correspond with the age effect. This should be very explicitly discussed that you make assumptions and in the discussion (and in the end of the abstract) you should be very careful with generalizing your interpreted results. |
| 98 | Five individual plots … -> not true, only for the pasture. This is misleading. |

| 100 | Your forests are quite small and your plots are taken as pseudoreplications within these small stands -> weakness of the study |
|---|---|
| 100 | Add also the area for the 40y old stand |
| 110 | Here it becomes clear that there are only 3 replications for the forest stands |
| 111 | I appreciate the effort to make soil profile pits to have a more detailed understanding of the belowground ecosystem (as well as a complete sampling of the roots) |
| 113 | Roots were counted on three profile walls -> per pit? Per replication? Unclear |
| 115 | In the end the results were pooled -> be already transparent about this in the M&M |
| 130 | Typo 45mm? So how many samples for the roots do you have per forest stand? 3 replications of pooled samples? In the figures it seems more? Unclear |
| 130 | Can you discuss the results of pooling these samples in the M&M section |
| 163 | I miss information about the density (do you have basal area)? |
| 194 | Not very clear "age gradient" in the results. This indicates that the differences between stands might just be random (site effects) that do not correspond with the age of the forest |
| 203 | between |
| 224 | You have a very high initial carbon stock -> could you explain more how come? |
| 228 | Careful with phrasing -> here you cannot say it is the effect of 40 years of afforestation for sure |
| 232 | Again very case specific |
| 247-248 | Can you really say it is forest age? |
| 305 | Here you speak of forest encroachment but you did not make this clear in the M&M. In that case how did the stand evolve? |
| 515 | Many outliers. Which ones are from the sides and which ones from the levels? Or are they pooled in that way as well. Be clear about the experimental design |

---

## Author Response (AR1)

Comments Reviewer 1

The manuscript shows the effects of a 130-year afforestation of alpine pasture on SOC stocks in the Swiss Alps. Authors found that afforestation did not necessarily change SOC stocks but clearly altered SOC dynamics. In general, the manuscript gives clear conclusions and has high scientific value. To improve the manuscript quality, the way of presenting data in figures, tables, and the Result section could be improved. In addition, the conclusion regarding words such as "increased", "no influence", and "moderately affected" can be connected with statistical significance (P values) in a more consistent and convincing way. Finally, several places showing inconsistent contents among figures & tables, results, and discussions should be addressed.

Thank you for the helpful comments and feedback. To improve the presentation of our figures, we added a map of the location of our study site within Switzerland and enlarged the points and font size for better visibility. We further replaced wording such as "moderately" and added statistical test results to our results We carefully checked the tables and figures for consistency among them.

Abstract:

Line 13-14 "SOC 20-year-old forest" and later

Thanks for pointing out the formatting inconsistency and we fixed it throughout the new version of the manuscript.

Line 22 "moderately affected the SOC stock"

I am not sure if "moderately affected" is a precise way to describe your results because SOC stocks remained constant in mineral soils but has a slight increase (P = 0.229) in organic horizons. When combining things together, I find a "slight increase in overall SOC stock" in line 15, a "moderately affected the SOC stock" in line 22, and "there is no SOC sequestration on ..." in line 25. I would suggest finding a better way to present them in order to avoid any potential confusion.

We agree that using terminology such as "moderately" or "slightly" is not precise. We removed the qualitative judgments and stated if a change is significant or not based on the results of the statistical analyses. We did this throughout the manuscript. (e.g., Line 12, 22)

Introduction:

It might be better to divide the Introduction into several paragraphs. For example, different paragraphs might focus on topics such as the significance of studies on relevant topics, the effects of afforestation on SOM, current knowledge gaps in the relevant study area, and so on.

We divided the introduction into individual paragraphs, which are organized by topic. Thanks for the suggestion.

Line 43-44 "OM accumulates in living biomass". It is a bit confusing to mention living biomass. Do you mean tree biomass? It could also be soil microbial biomass.

That is true that it could also be microbial biomass, so we changed "living biomass" into "tree biomass" (Line 46).

Line 75-76

To be prudent, it is suggested to specify "litter-derived carbon input" as something like aboveground or foliage litter because root litter is also a part of the litter.

We fully agree and changed it from "litter-derived carbon input" to "aboveground-derived carbon input" (Line 81).

Material and methods:

Line 115-123 and Line 143

How is SOC stock calculated? Is it corrected by the volume of stones? Based on the content of Line 115-123, it seems that soils collected by steel cylinders contained stones, which were removed later while passing through the 2mm sieve. I guess the bulk density should be corrected by stone volume. Is it possible to make it clearer?

Thank you for your questions. Yes, we corrected the SOC stock by the volume of the stones. We added this information in the M&M section (Line 127-128)

Line 119

Were bulk densities measured from organic samples? You calculated C stocks of organic horizons in the Result section.

Bulk densities for the organic horizon were not measured, as the samples were collected in an area of 0.25x0.25m without density cylinders and where then upscaled to m². We added this information in the M&M section (Line 130-131

Line 144-149

Based on Table 1, it seems the depth 0 cm started from the top of mineral soils. It looks like the formula only calculates SOC stock of the mineral soils. Were SOC stocks of organic horizons calculated using the same formula?

Indeed, we did not include the organic horizon in the mineral soil and therefore we calculated the SOC stock separately for the organic horizons and for the mineral soil. We now added the calculation of the SOC stocks of the organic horizons in the M&M section (Line 155-158).

Line 151-155

If I understand correctly, forest age and soil depth are both fixed effect predictors. Is this right? Where can I find the outcomes of mixed-effect models from Tables & figures and the Result section?

Correct, forest age and soil depth are modelled as fixed effects to test for their effect on SOC stocks. We added two additional tables in the supplementary material with the corresponding t- and  p-values  of the SOC and TN stocks (Table S4) and C:N ratio and

δ¹³C values (Table S5) as these are the variables we specify on in the discussion part of the manuscript.

Results:

3.1 and 3.2

It is not common that the results start with two sections based on supplementary materials. Maybe some information in the supplementary materials can be incorporated into figures and tables.

Thank you very much for the suggestion. We previously thought that this information is not so important, but based on your comment we realized that it would be beneficial to show this data in a figure. Therefore, we included the vegetation composition in the main text with a new figure (Figure 2).

Line 159 and later

It is a suggestion that it may be not necessary to present F values and P values at the same time whenever you make comparisons. Maybe only P value is enough; alternatively, even omit P values if you mention that the significant difference means $P < 0.05$. Instead, consider presenting significant differences in Figures and Table by adding P values or marking 'a', 'b', 'c'….

We think that it is important to show the statistics value (F-value) and what is the corresponding p-value. We therefore kept the F-value and p-value for completeness as the F-value, together with the theoretical value serves as a threshold to determine whether our samples are significantly different. We followed the recommendations from: https://link.springer.com/article/10.1007/s43440-020-00110-5

Line 159 and 167

Table S2 and Table S3?

We changed this accordingly in the text. Table S2 now refers to the composition of the Oi-horizon and Table S3 refers to the root biomass and root frequency.

Line 166

It seems that pinecones and branches do not match the names in Table S3. Branch = wood and twigs? Pinecones = spruce cones?

We corrected "pinecones" into "spruce cones" (Line 186) and changed this accordingly in the corresponding supplementary Table S2.

Line 175 and later

It is not necessary to mention P adj <0.95 all the time.

We removed this statement accordingly in the text.

Line185-188

The P value seems not to be 0.801 but (nearly-) significant if you look at the values of combined SCO stocks of organic horizons. (i.e. 1.7±0.2, 1.3±0.2, and 0.8±0.1).

This is because the statistical test was performed with the individual data points (n=27) and not with the sum of the final SOC stock. Only testing with the sum or average values would not be a representative result and there are to less arguments to perform a valuable statistical test. But we rephrased this on the text and make it clear that there is no statistical significant in the SOC stock within all organic horizons (Line 205 to 207), and then reported the sum (Oi, Oe, Oa) SOC stock of the organic horizon (Line 212-213).

Discussion:

Line 244 "… a significant lower SOC stock at 15 to 25 cm…"

Is it from Table 1? It will be easier for readers if significant differences could be found directly from tables and figures.

We apologize for the mistake. The lower SOC stocks are shown on Table 1, but they are not significantly lower for the 55-year-old forest. We thereby corrected this mistake (Line 271-272).

Paragraph 1 until Line 250:

I suggested making it clear that you are talking about mineral soils (next paragraph for organic horizons).

We made the differentiation between organic horizons and mineral soil clearer in the new version of the manuscript while referring to organic horizons and mineral soil, respectively when results are reported and / or discussed (Line 205 -213 SOC stock organic horizon, Line 213- 217 SOC stock mineral soil… – it is always mentioned in the beginning of the sentence if we report/discuss results of the organic horizons or of the mineral soil ).

Line 262 and after:

I think it might be helpful to make it clear when we can say something is non-significantly, significant, or unchanged. This does not mean that we need to draw a massive line between P < 0.05 and P > 0.05, but we should more or less be sure when we can say that A is higher than B. For example, it is not significant when P = 0.669 (line 245), but it is a (slightly) increased SOC sequestration when P = 0.229 (line 15, 195 and 262). I know that the latter might be a key conclusion for this manuscript, but it would be better to present it in a more convincing way.

Thank you for this comment, and we fully concur with this. We removed all qualitative statements and stated clearly if something is significant (p<0.05) or not. (Line 264; 272, 322)

Line 274 and after "…C:N ratios of Oi horizons increased…"

Table 1 should be Table 3?

Thanks for pointing this out and we fixed this typo to Table 3.

Please be aware in lines 203-204 you said C:N ratios were not significantly different between forest ages, but you mentioned here C:N ratios increased with age. Values of 42.4±10.8 indicated a very large variation within the samples of 130-year-old soils. This is probably the reason why it is not statistically significant.

Thank you for your comment. The reason why we stated it in this way was because the C:N ratio of the Oi horizon is variable because of the number of branches, spruce cones, needles, etc. yet and despite this variation they do not significantly differ with age. We made it now clearer that despite this apparent increase in C:N ratio at the 130-year-soils in comparison to younger forest stand ages, for our sample this is not significantly different (Line 301-302).

Line 287 and after

In general, organic matter input from the root is comprised of root litter and root exudates, from which the exudate is considered as the more important belowground input (Sokol et al. 2018 doi: 10.1111/nph.15361). Root exudates contain a lot of easily-decomposable (high-quality) compounds, which promote the microbial transformation of these compounds and the formation of microbial-derived compounds as stabilized SOC with low C:N ratios (Cotrufo et al. 2013, you cited). It is possible that the intensive (or effective) utilization of root exudates promotes the formation of stable microbial-derived organic matter, and therefore, the C:N ratios were still low. Therefore, my suggestion is that try to discuss it in a more convincing way with the consideration of the transformation of root exudates

Thank you for suggestion of this potential mechanism and we included this possibility in the discussion (Line 314-316).

Figure and Table:

In general, there is no information of significant difference in figures and table. This makes it difficult to link figures and tables to the Result and Discussion sections. Please consider presenting significant differences in Figures and Table by adding P values or marking 'a', 'b', 'c'….

We included significance indications (*, **, ***, and n.s.) in the corresponding figures.

Table 1

When combined organic and mineral soils, SOC was 11.6 + 0.8 =12.4 for 40-year-old soil and 11.0 + 1.3 = 12.3 for 130-year soil, respectively. In line 194, the 40-year-old soil and 130-year-old soil had 7% and 11% higher SOC stocks than pasture soil, respectively. Is this a mistake?  I think both organic and mineral soils should be included when compare total SOC stocks.

Thank you for pointing this out and we corrected this in the new version in the manuscript (Line 217-218)

Table 2

63.5 + 02.8, remove '0'.

Apologies: we removed the 0 in Table 2.

Comments Reviewer 2

The study presented here covers a topic that is interesting and relevant since many pastures in alpine regions are abandoned and expected to become forests in the future – which could feed back to climate change. Knowing the impact of this land-use conversion is needed. The topic and proposed research questions however are not very novel and do not contribute much new knowledge.

Thank you for your careful assessment of our manuscript. We are aware that we work at a single site planted with a single tree species is case specific. However, such forest stands are still very common in subalpine and alpine settings, and in particular for steep slopes they serve as a nature-based solution for avalanche protection and slope stabilization. Further, while the questions might not be novel in general, we believe they become interesting because our study site is very unique in its long-term afforestation chrono-sequence with four forest stands of different age classes and a reference pasture area on a rather homogenous slope. Moreover, it provides further information about the subalpine zone, as studies concerning the organic carbon stock in the Alps are scarce (Thuille and Schulze, 2006; Zimmermann et al., 2010) and there is an ongoing process of subalpine pasture abandonment and forest encroachment, which policy makers consider as beneficial in terms of carbon storage. However, recent studies (Clemmensen et al., 2015; Friggens et al., 2020) show evidence that priming, mycorrhizal nutrient mining associated with ectomycorrhizal tree species can result in a net carbon loss from the mineral soil. We therefore used this afforestation chrono-sequence as a natural experiment to test whether there is an increased carbon sequestration with increasing forest age. We tried to better highlight the benefits of our approach and novelty of our findings in the updated version of the manuscript.

| Line | Comment |
|---|---|
| 19-22 | As roots … -> unclear sentence
We changed this sentence accordingly: " Due to the high root C:N ratio (pasture 63.5 ± 2.8 and forests between 54.7 ± 3.9 and 61.2 ± 2.9), particulate root-derived organic matter seems to have a rather small effect on the forest soil C:N ratio as well as on the SOC accumulation in the mineral soil " (Line 19-22) |
| 22-25 | Is this novel enough?
Although there are several studies available on similar topics, still the alteration in the SOM dynamics following afforestation in subalpine settings is not fully understood, as it varies among locations, depending on soil type, tree species, climate conditions etc. Consequently, our study is one of very few studies on SOM dynamic following afforestation with coniferous trees in alpine ecosystems. |
| 37-38 | Afforestation is a promising measure -> is this also true for alpine regions where albedo effects might have a negative impact on climate change
Thank you for your comment. Yes, afforestation is still a promising strategy also in alpine regions, as afforestation still has the potential to increase SOC stocks (although context-dependent) and because forest areas are important in alpine regions as they have a surface cooling effect in summer compared to open land (Tang et al., 2018; Sofiadis et al., 2022) which results in less soil surface warming and less SOC losses. But we are aware of the albedo effect, which we did not include in this study as it was beyond the scope of our study. |
| 49-50 | Be explicit and call it "context-dependency"
Thank you for the suggestion, we changed this, accordingly (Line 52). |
| 51 | Initial carbon stock is a key factor -> yet you do not take this into account in your research
That is correct, and we removed this statement in the new version of the manuscript (Line 53-55) |
| 54-55 | Transition line 54 to 55 not very fluent (10-20 cm is not that deep? Was that the point?)
We removed this qualifier of "deep soil layers" (Line 57). |

| | |
|---|---|
| 58-59 | Age is underexplored -> I think this is the main strength of your study that you have a large gradient in time going back 130 years
Thank you for your assessment and we add this information more explicitly in the introduction and again in the conclusion to pointed out that one strength of this study is the long-term duration of the afforestation sequence.( Line 63-64; Line 333). |
| 75-77 | Clear hypothesis. Is (iii) really needed? It is accepted as common knowledge I would say
Thanks for your suggestion. We do agree that in general an increase in SOC stock in the organic horizons is common knowledge. We therefore combined hypothesis ii and iii in "hypothesis ii". |
| Introduction | Overall very well written introduction with nice literature review.
Thank you very much! |
| 85 | Afforested -> Encroached? Natural regeneration? Planted to avoid erosion? Not explicitly stated in the M&M. If planted, at what planting distances? What is was the basal area (the density) of the stands at the time of sampling. This is important information that is needed to interpret the results
In our case, afforestation was due to plantations in the early 1950s as protection against avalanches. We stated in the M&M section that our study site is a human-induced afforestation chrono-sequence (Line 92).The tree density is 58 ± 5% in the 40-year-old forest, 48 ± 3% in the 55-year-old forest, and 65 ± 5% in the 130-year-old forest. The tree density (canopy cover) is now shown in a separate figure and included in the main text (Figure 2). |
| M&M | There are no replications of the different ages. The replications are made within the same forest stands resulting in pseudoreplications. This means you cannot say anything about the effect of age / time since afforestation. It can just be differences in between individual stands that do not correspond with the age effect. This should be very explicitly discussed that you make assumptions and in the discussion (and in the end of the abstract) you should be very careful with generalizing your interpreted results.
We fully agree with your point and that we do have repeated measures of the same stand instead of repeated measures of multiple stands of the same age. We therefore, removed "general statements" about our results in the new version of the manuscript. Furthermore, we more carefully rephrased some of our conclusions (e.g., Line 21-25; 313-316). |
| 98 | Five individual plots … -> not true, only for the pasture. This is misleading.
Thanks for pointing this out, we made this clearer in the M&M section in the revised version of the manuscript (Line 109-110; 115-116, 121-122). The number of trees as well as the canopy cover were determined for all five plots in forest areas, but soil samples and organic horizons were collected for three plots in the forest, because of time constraints during the sampling campaign. We marked the plots in Figure 1 where mineral soil and organic horizons were collected and added this information in the caption of the corresponding figure. |
| 100 | Your forests are quite small and your plots are taken as pseudoreplications within these small stands -> weakness of the study
Thank you for your comment. Indeed, these are important limitations of our study. However, based on observations made by Hiltbrunner et al. (2013), we could ensure comparable slopes and soil types in the area, which often cannot be ensured in alpine and subalpine settings because of the heterogeneity of the slope and historical developments such as glacier retreats. Therefore, our sample site still has certain strengths that issues such as heterogeneity of various environmental and soil related factors can be neglected. Therefore, we believe that the results still provide interesting reference data on these types of locations over a long afforestation period and advance interesting new hypotheses to be tested further. For example, is there really a peak in SOC at mid-afforestation period? What do these variations within stand tell us about the different soil stocks? We believe that descriptive and site-specific studies still have an added value also to contribute to larger initiatives as for example larger databases, etc. |
| 100 | Add also the area for the 40y old stand
We included the area of the 40-year-old forest (0.55 ha; Line 105). |
| 110 | Here it becomes clear that there are only 3 replications for the forest stands
We made this clear in the M&M section in the new version of the manuscript (Line 115-116) |

| | |
|---|---|
| 111 | I appreciate the effort to make soil profile pits to have a more detailed understanding of the belowground ecosystem (as well as a complete sampling of the roots)
Thanks. |
| 113 | Roots were counted on three profile walls -> per pit? Per replication? Unclear
The roots were counted in each soil pit on the back wall, on the left, and on the right wall. To be consistent with the soil sampling, we have 5 replicates in the pasture and 3 replicates in each forest stand age. We included this information in the M&M section (Line 121-123). |
| 115 | In the end the results were pooled -> be already transparent about this in the M&M
We included this information in the M&M section (Line 134-138) |
| 130 | Typo 45mm? So how many samples for the roots do you have per forest stand? 3 replications of pooled samples? In the figures it seems more? Unclear
Roots were combined as the number of roots < 2mm was too small to mill and to analyse. The number of root samples (0-5mm) was 40 in the pasture, 25 in the 40-year-old forest, 25 in the 55-year-old forest, and 26 in the 130-year-old forest. The number of replicates is the number of soil pits, e.g., 5 in pasture and 3 in each forest stand. The variability in the number of root samples represents the natural variability in the field with some soil horizons having more, some having less roots. We clarified this in the M&M section and included the number of fine and coarse roots for pasture and forest stands (Line 134-138) |
| 130 | Can you discuss the results of pooling these samples in the M&M section
Combining the root samples (0-5mm) gives a more representative as well as a consistent and comparable result between these two ecosystems. We tested an amount of fine and coarse root of the same sample for statistical difference between carbon and nitrogen content without a significant difference, thereby we expect that this pooling has not much effect on our results. But we added this information in the M&M section and explained why we decided to pool these samples (Line 136-138). |
| 163 | I miss information about the density (do you have basal area)?
Thank you for your comment. We added this information in the M&M section (Line 110) as the number of trees and the canopy cover was obtained in a squared area of 10m x 10m at each plot with the soil pit located in its centre, if a soil pit was prepared. |
| 194 | Not very clear "age gradient" in the results. This indicates that the differences between stands might just be random (site effects) that do not correspond with the age of the forest
Thank you for explaining your interpretation. Our interpretation is slightly more nuanced. Indeed, when considering the carbon stock in the mineral soil, we find no differences between soils of each forest stand age and the pasture. But for the C stock of the organic horizons, we find that forest age 55 has a significantly higher stock than forest age 40. |
| 203 | between
Thanks for finding this typo, we have fixed it. |
| 224 | You have a very high initial carbon stock -> could you explain more how come?

Actually, this is not a very high initial carbon stock for alpine areas. We added an additional reference who reported similar SOC stocks for alpine pastures as we reported in our study, and added information why the SOC stock reported by Zeeman et al (2010) could be lower than the ones reported in our study (Line 250-254). |
| 228 | Careful with phrasing -> here you cannot say it is the effect of 40 years of afforestation for sure
Thank you for mentioning this. Of course, there could be other factors contributing to the increase of SOC following afforestation beyond the establishment of trees. It is often discussed in the literature, and the majority of the afforestation studies found an increase in SOC stocks 30 to 40 years after afforestation (Thuille and Schulze, 2006; Hiltbrunner et al., 2013). We thereby left this statement that there is an increased SOC stock after 40 years in the discussion, but we did not imply that this is a direct effect of afforestation (Line 256-258) |
| 232 | Again very case specific |

| | |
|---|---|
| | We do agree that this is specific to our forest areas, and we will discuss it in context of what we know about alpine and temperate forests. (Line 260-261) |
| 247-248 | Can you really say it is forest age?
We believe so because other parameters in this study site had no influence in the SOC stock, e.g., slope or erosion (Hiltbrunner et al., 2013) or e.g., soil type, soil pH values, bulk densities (etc.). As such we are confident in saying that additional ten years might not increase SOC stocks. We rephrased this in the new version of the manuscript and made clearer that this might be one explanation for the unchanging SOC stock (Line 273-275). |
| 305 | Here you speak of forest encroachment but you did not make this clear in the M&M. In that case how did the stand evolve?
Thank you for bringing this up; indeed, we used the terms interchangeably and have corrected this in the new version of the manuscript. We now use afforestation to keep consistency (Line 332) |
| 515 | Many outliers. Which ones are from the sides and which ones from the levels? Or are they pooled in that way as well. Be clear about the experimental design
They are not pooled. The results do not refer to the roots counted in the field (root frequency), but the root biomass obtained from the volumetric cylinders. |

There are two things I value in the current manuscript:

- The long gradient in time (if the chronosequence would have been replicated) adding more information on the long term impacts of afforestation

- The study is well executed (- the design) and the manuscript is well written

Thank you.

My major concern:

- The study is based on pseudoreplications so the conclusions in terms of time since afforestation are not supported by the data, they could be coincidence or site effects. In my opinion this is a major issue that is hard to resolve.

We do understand your point about "pseudoreplication" and we are aware of these limitations in our experimental design and, as you state, these are difficult to resolve. However, the study site is characterized by a homogenous soil depth and carbon distributions across the slope (Hiltbrunner et al., 2013) and thus, we are confident that the detected changes therein should be caused by the different forest ages. Our sampling protocol further ensured a minimum distance of ~5m and a minimum elevation lapse of 5 to 10m between individual plots of the same forest age class. Given the heterogeneity of soil properties and processes therein (Hiltbrunner et al., 2013) we think that the independence of 3 to 5 observations we made within the individual forest ages, can be used as true statistical replicates, rather than pseudoreplicates. Further, in light of what we know, these small scale and local studies can contribute with reporting results that are local and fundamental to studies such as meta-analyses.

Specific comments:

Literature

Clemmensen, K. E., Finlay, R. D., Dahlberg, A., Stenlid, J., Wardle, D. A., and Lindahl, B. D. (2015). Carbon sequestration is related to mycorrhizal fungal community shifts during long-term succession in boreal forests. New Phytologist, 205, 1525-1536. https://doi.org/10.1111/nph.13208

Friggens, N. L., Hester, A. J., Mitchell, R. J., Parker, T. C., Subke, J. A., and Wookey, P. A. (2020). Tree planting in organic soils does not result in net carbon sequestration on decadal timescales. Global Change Biology, 26, 5178-5188. https://doi.org/10.1111/gcb.15229

Hiltbrunner, D., Zimmermann, S., and Hagedorn, F. (2013). Afforestation with Norway spruce on a subalpine pasture alters carbon dynamics but only moderately affects soil carbon storage. Biogeochemistry, 115, 251-266. https://doi.org/10.1007/s10533-013-9832-6

Sofiadis, G., Katragkou, E., Davin, E. L., Rechid, D., de Noblet-Ducoudre, N., Breil, M., Cardoso, R. M., Hoffmann, P., Jach, L., Meier, R., Mooney, P. A., Soares, P. M. M., Strada, S., Tölle, M. H., and Warrach Sagi, K. (2022). Afforestation impact on soil temperature in regional climate model simulations over Europe, Geoscience. Model Dev., 15, 595–616, https://doi.org/10.5194/gmd-15-595-2022

Thuille, A., and Schulze, E. D. (2006). Carbon dynamics in successional and afforested spruce stands in Thuringia and the Alps. Global Change Biology, 12, 325-342. https://doi.org/10.1111/j.1365-2486.2005.01078.x

Tang, B., Zhao, X., and Zhao, W. (2018). Local effects of forests on temperatures across Europe. Remote Sensing, 10, 529. https://doi.org/10.3390/rs10040529

Zimmermann, P., Tasser, E., Leitinger, G., and Tappeiner, U., 2010. Effects of land-use and land-cover pattern on landscape-scale biodiversity in the European Alps. Agriculture, Ecosystems and Environment 139, 13–22. https://doi.org/10.1016/j.agee.2010.06.010